# Nonsequential Splicing Events Alter Antisense-Mediated Exon Skipping Outcome in *COL7A1*

**DOI:** 10.3390/ijms21207705

**Published:** 2020-10-18

**Authors:** Kristin A. Ham, May Thandar Aung-Htut, Sue Fletcher, Steve D. Wilton

**Affiliations:** 1Centre for Molecular Medicine and Innovative Therapeutics, Health Futures Institute, Murdoch University, Murdoch 6150, Australia; Kristin.Ham@murdoch.edu.au (K.A.H.); M.Aung-Htut@murdoch.edu.au (M.T.A.-H.); s.fletcher@murdoch.edu.au (S.F.); 2Perron Institute for Neurological and Translational Science, Centre for Neuromuscular and Neurological Disorders, The University of Western Australia, Nedlands 6009, Australia

**Keywords:** *COL7A1*, dystrophic epidermolysis bullosa, antisense oligonucleotides, splice-switching, exon skipping, intron retention, morpholino, 2′-*O*-methyl

## Abstract

The *COL7A1* gene encodes homotrimer fibrils essential for anchoring dermal and epidermal layers, and pathogenic mutations in *COL7A1* can cause recessive or dominant dystrophic epidermolysis bullosa. As a monogenic disease gene, *COL7A1* constitutes a potential target for antisense oligomer-mediated exon skipping, a therapy applicable to a growing number of other genetic disorders. However, certain characteristics of *COL7A1*: many exons, low average intron size, and repetitive and guanine-cytosine rich coding sequence, present challenges to the design of specific and effective antisense oligomers. While targeting *COL7A1* exons 10 and 73 for excision from the mature mRNA, we discovered that antisense oligomers comprised of 2′-*O*-methyl modified bases on a phosphorothioate backbone and phosphorodiamidate morpholino oligomers produced similar, but distinctive, splicing patterns including excision of adjacent nontargeted exons and/or retention of nearby introns in some transcripts. We found that the nonsequential splicing of certain introns may alter pre-mRNA processing during antisense oligomer-mediated exon skipping and, therefore, additional studies are required to determine if the order of intron removal influences multiexon skipping and/or intron retention in processing of the *COL7A1* pre-mRNA.

## 1. Introduction

Dystrophic epidermolysis bullosa (DEB) is an inherited disease caused by mutations in *COL7A1* that compromise expression of functional gene products. Dystrophic epidermolysis bullosa can be inherited in either a dominant (DDEB, Phenotype MIM number 131750) or recessive (RDEB, Phenotype MIM number 226600) manner, with the latter generally resulting in the most severe disease. Mutations that result in premature termination codons, affecting both alleles, and resulting in absent or markedly reduced type VII collagen protein (C7), are characteristic of RDEB, termed generalized severe [1]. These mutations can result in reduced *COL7A1* mRNA through nonsense-mediated decay, and absent or incomplete polypeptides, leading to lack of anchoring fibril deposition in the epidermal basement membrane layers. A milder recessive form, RDEB, termed generalized intermediate, is commonly caused by compound heterozygous mutations whereby one *COL7A1* allele leads to premature termination of translation and the other carries a missense mutation, frequently a glycine substitution. Full-length C7 polypeptide is still produced, albeit the missense mutations compromise the conformation, structure and, therefore, stabilization of the anchoring fibrils, by creating a kink or abnormal folding that disrupts the normal trimerization of the C7 propeptides. The *COL7A1* gene is distinct from other collagen genes in that some glycine substitutions can be silent in the heterozygous state, while other glycine substitutions may manifest in the generally milder, dominantly inherited dystrophic epidermolysis bullosa. Current therapies provide some symptomatic relief but do not address the underlying absence of C7 [2]. Therefore, there is an urgent need to develop effective therapies for this devastating disorder.

The *COL7A1* gene is of average length, spanning 31.2 kb, with the 118 exons in the 9.5 kb mature mRNA encoding collagen VII peptides that ultimately assemble into anchoring fibrils. Mutations that compromise normal expression of this gene result in severe blistering of the skin and mucous membranes. The 8.83 kb coding sequence for *COL7A1* exceeds the capacity for delivery by current single adeno-associated virus vectors, but administration through the dual vector adeno-associated virus systems is being explored [3]. Considering the nature of the gene and gene product, alternative therapeutic strategies such as antisense oligomer (AO)-induced exon skipping are being considered.

Of the 118 exons in *COL7A1*, 107 are in-frame, and specific removal of one of these exons would not disrupt the reading frame. The triple-helical domain is characterized by a repetitive glycine-X-Y amino acid sequence, where X and Y can be any amino acid [4]. The number of repeats does not appear to impact triple helix formation, but conserving the glycine-X-Y repeat pattern and the carboxyl-terminal domain is essential for correct homotrimer folding and C7 protein functionality [5]. Once secreted from the cell, the carboxyl-terminal non-collagenous 2 domain of the C7 homotrimer is cleaved, and the mature homotrimers assemble into antiparallel dimers stabilized by disulfide bridges to form important anchoring fibrils that connect the epidermal basement membrane layers [6,7,8].

Exon 73 was chosen for initial study as it contains the highest number of reported mutations [9,10], and exclusion of this exon is reported to have no significant impact on the triple helix formation or stability of the protein [11]. Exon 10 was also targeted to compare the feasibility of promoting AO-induced splice-switching in a region of the *COL7A1* pre-mRNA transcript encoding a non-collagenous domain. Here, we report altered splicing of exon 10 and 73 during processing of the *COL7A1* pre-mRNA, whereby very subtle changes in the AO annealing site and chemistry induced distinct mRNA splice isoforms. We speculate that since *COL7A1* exons are separated by small introns, targeting a single exon may impact on the splicing of adjacent exons, with considerable cross-talk between these exons. Importantly, some of these multiexon skipping and intron retention events would not be detected with reverse transcription polymerase chain reaction (RT-PCR) assays designed to detect excision of the targeted exon, highlighting the need to include several flanking exons in any RT-PCR assay designed to monitor changes in pre-mRNA processing.

For both exons 10 and 73 we show that the phosphorodiamidate morpholino oligomer (PMO) chemistry was able to enhance exon skipping in comparison to that induced by a corresponding oligonucleotide composed of 2′-*O*-methyl modified bases on a phosphorothioate backbone (2′-OMe) using the methods tested. We also encountered some challenges while seeking to develop AO-mediated therapeutics for RDEB, including designing and optimizing AO candidates specific to the target without causing mis-splicing events in other regions of this highly repetitive coding sequence. We sought to study the mechanisms behind the observed mis-splicing events and found that the nonsequential splicing of certain introns may alter pre-mRNA processing during AO-mediated exon skipping.

## 2. Results

### 2.1. COL7A1 Exon 10

Three 2′-OMe AOs were designed to target the exon 10 acceptor and donor canonical splice sites, and intraexonic predicted exon splice enhancers (ESE; Figure 1a). Oligomers H10A(+65+85) and H10D(+07-16) targeting intra-exonic ESE motifs and the donor splice site, respectively, both resulted in 34% of the transcripts missing exon 10 after transfection at 50 nM concentration, with a dose-response of 26% or 19% exon 10 exclusion after transfection at 10 nM, respectively. The 2′-OMe AO targeting the acceptor splice site H10A(-05+20) did not alter *COL7A1* transcript processing at any concentration tested, in essence serving as an unintended negative control (Appendix A). The combination of H10A(+65+85) and H10D(+07-16), when transfected at 50 nM (25 nM each), increased levels of exon 10 skipped transcripts to 64%, while transfection at 10 nM (5 nM each) caused skipping of exon 10 in 48% of transcripts (Figure 1b). No other transcript variants were detected after RT-PCR spanning exons 8 to 13, indicating specific skipping of the targeted exon. Oligomer sequences H10A(+65+85) and H10D(+07-16) were evaluated as PMOs, introduced into cells by nucleofection at a concentration of 150 µM in the cuvette, and incubated for four days before RT-PCR analysis. Robust exon 10 removal was induced by the PMO chemistry, including by transfection of a cocktail combining both compounds (Figure 1c). Treatment with H10A(+65+85) resulted in 10% full-length transcripts, approximately 50% of the transcripts missing exon 10, with the remainder of the products being derived from multiple aberrant splicing events: full-length plus intron 9 and 10; full-length plus intron 9; and exons 9 and 10 removed from the transcript, all confirmed by sequencing (Figure 1c,d). The largest amplicon, approximately 950 nucleotides in length, was consistent in size with the full-length product but with retention of introns 8, 9 and 10. Treatment with H10D(+07-16) was more effective and specific in promoting exon 10 removal, with approximately 95% of transcript products missing exon 10, and other transcripts representing 5% of total amplicons (full-length amplicon plus introns 9 and 10 as confirmed by Sanger sequencing and the presumed full-length plus introns 9, 10, and 11, not confirmed by Sanger sequencing; Figure 1c,d). Combination of both PMO oligomers in a cocktail, transfected at 150 µM (75 µM each) elicited robust exon 10 removal but also resulted in 16% aberrant transcripts (amplicons missing exons 9 and 10; Figure 1c,d).

### 2.2. COL7A1 Exon 73

The preliminary evaluation of 2′-OMe AOs targeted to *COL7A1* exon 73 was performed with 25mers, and variable levels of exon skipping were induced with AOs H73A(-10+15), H73A(+16+40), and H73A(+41+65), but not with H73A(+66+90) (Figure 2b). Oligomers targeting motifs further downstream of the H73A(+66+90) annealing site were completely ineffective in modifying *COL7A1* transcript splicing under these transfection conditions (Appendix A). Sanger sequencing was performed on anticipated and unexpected amplicons, and these were confirmed to result from skipping exon 73, excising exon 73 but retaining intron 76, missing exons 73 and 74, or skipping exons 73 and 74 and including intron 76 (Figure 2e). Since the oligomer that skipped exon 73 most efficiently, H73A(+16+40), could potentially anneal to other exons, attempts were made to increase specificity for exon 73 by reducing the AO length by five nucleotides from either the 5′ or 3′ end. Removing bases from the 5′ end of H73A(+16+40) to produce H73A(+16+35), did not abrogate intron 76 inclusion, whereas removing five nucleotides from the 3′ end to generate H73A(+21+40), enhanced exon skipping specificity and reduced both intron 76 inclusion and exon 74 skipping (Figure 2c). The 2′-OMe AO, H73A(+21+40), removed exon 73 specifically from approximately half of the transcripts at the highest concentration tested (Figure 2c). Additional oligomers were designed to target the ESE site, and we found that any oligomer covering a motif predicted to be a target for serine and arginine rich splicing factor (SRSF) 5, even by two nucleotides (Figure 2a), resulted in exon 74 skipping and intron 76 inclusion (Figure 2c,e).

Oligomer sequences H73A(+16+40), H73A(+16+35) and H73A(+21+40) were subsequently ordered as PMOs, nucleofected into cells and incubated for four days before RNA extraction and analysis. Nucleofection with the 25mer PMO H73A(+16+40) induced a similar pattern of *COL7A1* transcript products observed when cells were treated with the equivalent 2′-OMe AO, with some transcripts missing exon 73 (22%), exons 73 and 74 (51%), or missing both exons, but with the retention of intron 76 (28%) (Figure 2d). However, both 20mer PMOs were more specific, predominantly removing exon 73, with some skipping of exon 74 (Figure 2d). Intron 76 is 75 nucleotides in length and, although inclusion of this intron would not cause a frameshift, inclusion of an in-frame termination UGA 17 codons into the intron (Figure 2e) would result in premature termination of translation and render this transcript susceptible to nonsense-mediated decay. 

### 2.3. Splicing Order Analysis

The order of intron removal in many genes is not necessarily sequential and, based on the results from AO-targeted exon 10 and 73 skipping, this may be the case for *COL7A1* pre-mRNA processing. Therefore, we further investigated the order of intron removal from introns 72 to 76 in the *COL7A1* pre-mRNA. A pairwise comparison of *COL7A1* from introns 72 to 76 was performed by adopting a method developed by Kessler et al. [13] to determine the order of intron removal, using strategically placed primer sets to amplify the pre-mRNA molecule. Briefly, two sets of primers were used to determine the order of splicing of two adjacent introns. Each primer set contained an exon and an intron primer to preferentially amplify the pre-mRNA transcripts and minimalize any contribution from the mature mRNA. This method was repeated for all adjacent introns between exons 72 and 77 in *COL7A1* mRNA (primer sets A-H), and then repeated for every second intron (primer sets I-N) (Figure 3a). Genomic DNA was amplified to represent the expected size of the unspliced transcript product. The no-reverse transcription control was included to confirm that the RNA sample used for analysis was free from DNA contamination. The order of intron removal order is shown in Figure 3b and summarized in Table 1.

**Intron 72 versus intron 73.** Primers that extend from exon 72 (primer 1) to intron 73 (primer 9) generate the spliced product primarily, with a hint of unspliced product, indicating that intron 72 is usually removed before intron 73. The unspliced product predominates over the spliced counterpart when amplified using primers extending from intron 72 (primer 2) to exon 74 (primer 10), indicating that intron 73 is usually removed after intron 72.**Intron 73 versus intron 74.** Primers that extend from exon 73 (primer 3) to intron 74 (primer 11) yield mainly the unspliced product, with traces of the spliced product, indicating intron 74 is usually removed before intron 73. Primers that extend from intron 73 (primer 4) to exon 75 (primer 12) again yield mainly the unspliced product, with a hint of the spliced product, indicating intron 73 is usually removed before intron 74. These results indicate no preference to the order of removal of introns 73 and 74.**Intron 74 versus intron 75.** Primers that extend from exon 74 (primer 5) to intron 75 (primer 13) yield the unspliced product, indicating intron 75 is usually removed before intron 74. Primers that extend from intron 74 (primer 6) to exon 76 (primer 14) again yield the unspliced product and the spliced product, indicating intron 75 is removed before intron 74.**Intron 75 versus intron 76.** Primers that extend from exon 75 (primer 7) to intron 76 (primer 15) yield both the unspliced and spliced transcripts, indicating intron 75 is removed before intron 76. Primers that extend from intron 75 (primer 8) to exon 76 (primer 16) yield solely the unspliced product, suggesting intron 75 is removed before intron 76.**Intron 72 versus intron 74.** Primers that extend from exon 72 (primer 1) to intron 74 (primer 11) yield mainly the unspliced products and transcripts missing intron 72 are a minor product. Primers that extend from intron 72 (primer 2) to exon 75 (primer 12) yield the unspliced product, and possibly transcripts either missing intron 74 or missing intron 73 + 74. Since a similar-sized product is also present in the genomic DNA amplification for this primer set, we cannot determine if intron 73 or 74 splicing occurred. The results from this primer set are, therefore, inconclusive but are suggestive of no preference to the order of removal of these introns.**Intron 73 versus intron 75.** Primers that extend from exon 73 (primer 3) to intron 75 (primer 13) yield mainly the unspliced product. Transcripts missing intron 73 or 74 individually, and transcripts missing introns 73 + 74, are minor products. Primers that extend from intron 73 (primer 4) to exon 76 (primer 14) yield mainly the unspliced product, transcripts missing intron 75, and transcripts missing both introns 74 + 75. Transcripts missing intron 74 is a minor product. Since the level of intron 75 spliced product is comparable to that of the unspliced molecule, intron 75 tends to be removed before intron 73.**Intron 74 versus intron 76.** Primers that extend from exon 74 (primer 5) to intron 76 (primer 15) yield mainly transcripts missing intron 75, and transcripts missing introns 74 + 75, with the unspliced product present as a minor product. Primers that extend from intron 74 (primer 6) to exon 77 (primer 16) yield primarily the unspliced product and the product missing intron 75. However, no products missing intron 76 are observed. This splicing pattern indicates intron 75, and a combination of introns 74 + 75 are spliced before intron 76.

In summary, the data suggest that introns 72 and 75 are removed before introns 73 and 74, which are in turn removed before intron 76.

## 3. Discussion

The therapeutic potential of antisense-mediated splice-switching is gaining recognition, beginning with the accelerated approvals of Eteplirsen and Golodirsen to treat the most common mutation subgroups that cause Duchenne muscular dystrophy (MD), coupled with the approval of Spinraza as a therapy for spinal muscular atrophy [14,15,16]. In Duchenne MD, mutations that lead to premature termination of translation are associated with severe disease, whereas most in-frame deletions are associated with the milder allelic condition, Becker MD. As is the case with many Duchenne MD-causing mutations in the dystrophin gene, severe disease-causing gene lesions in *COL7A1* could be amenable to a targeted exon skipping therapy. We have considerable experience in applying and developing splice-modulating therapies to treat Duchenne MD and sought to apply this technology to other amenable conditions. Hence, targeted exon skipping to remove disease-causing *COL7A1* exons carrying premature translation termination mutations was investigated as a potential treatment for RDEB.

Although mechanisms of antisense-mediated exon skipping in the *COL7A1* transcript may follow the same fundamental concepts as targeted splicing interventions in dystrophin gene transcripts, there are some major differences between these genes, and challenges that might influence the efficacy or specificity of the process. Despite a mature mRNA of 14 kb, the dystrophin gene is recognized as the largest in the human genome with 79 exons spanning more than 2.3 Mb. In contrast, the very compact *COL7A1* gene consists of 118 exons spread across 31.2 kb and is of a similar length to the average human gene that consists of 8 or 9 exons processed into approximately one kb of mRNA [17]. Consequently, neither *DMD* or *COL7A1* may be regarded as averages, but extremes that lie at either end of gene structure and organization. The *COL7A1* gene is studded with many short introns, with an average length of 188 nucleotides, compared to the *DMD* introns, 17 of which are longer than the entire *COL7A1* [4,18]. As such, one could expect substantial differences in AO design for efficient splice-switching, especially taking into account the guanine-cytosine (GC)-content of exons and introns, as well as the temporal and tissue-specific issues of cotranscription and splicing.

Intron 30 in *COL7A1* is the smallest, measuring 70 nucleotides in length and is theoretically too small for the conventional spliceosomal A complex, which is reported to bind the entire length of a 79 to 125 nucleotide single-stranded RNA sequence [19]. The shortest dystrophin intron, 14, is 107 bases long, and only two other dystrophin introns are less than 500 nucleotides in length. In contrast, merely seven out of the 117 *COL7A1* introns are longer than 500 residues, highlighting the major discrepancy in exon density between these mature mRNAs. Dystrophin transcription across all 79 exons has been estimated to take approximately 16 h [18,20], compared to *COL7A1*, where transcription should take less than 13 min based upon an average elongation rate of 2.4 kb/min [21]. Consequently, where it may take less than 30 s to transcribe several *COL7A1* exons and introns, transcription across dystrophin exons 43 to 46 should take nearly 150 min. Indeed, when first embarking upon targeted exon skipping in the *COL7A1* transcript, we were uncertain if cotranscriptional splicing would be so tightly linked that the AOs would not have time to anneal and mask the exon for exclusion from the mature mRNA.

We had anticipated that RT-PCR across the *COL7A1* transcript could be challenging due to the repetitive nature of the collagenous coding sequence, as well as the high GC-content of 64% across the coding region. The collagenous region, a repeating pattern of glycine-X-Y amino acids [4], spans from exons 29 to 112 and is comprised of small exons of typically 27 to 81 nucleotides in length (only five exons were larger than 81 nucleotides: 96, 108, 117, 123 and 201). However, despite the high GC-content and repetitive nature of the sequence, RT-PCR amplification was surprisingly efficient and specific once a high-fidelity GC-rich PCR system was employed. Several minor, unexpected products were observed in both AO-treated and untreated samples during the generation of some amplicon sets. However, the high GC-content of the entire amplicon from exons 8 to 13 (63%) and exons 72 to 77 (68%) could explain this result. Templates higher than 60% GC-content are considered GC-rich [22] and can form secondary intramolecular structures such as hairpins that can disrupt *Taq* polymerases and inhibit efficient PCR amplification [23]. Another explanation is imperfect splicing in the region, whereby some introns are not removed as efficiently as others and some transcripts remain incompletely processed. 

The repetitive nature of the coding domain sequence raised the possibility that some oligonucleotides could anneal to either distal or proximal homologous sequences. Most splice-switching AOs designed for *DMD* exon skipping are between 25 and 30 nucleotides in length, while the GC-rich *COL7A1* exons encoding the repeated glycine-X-Y motif would suggest that the possibility of cross hybridization to homologous sequences would be high. The three exon 73 AOs found to influence the processing of the *COL7A1* transcript, the 25mer H73A(+16+40) and 20mers H73A(+16+35) and H73A(+21+40), were analyzed using the search engine Basic Local Alignment Search Tool (BLAST) [24]. All three oligomers were predicted to anneal to several other regions in the *COL7A1* pre-mRNA, with a maximum of 10 consecutive nucleotides annealing in exon 4, highlighting the difficulty of designing oligonucleotides to a specific target in a highly repetitive gene transcript such as *COL7A1*. Nevertheless, RT-PCR across exons 2-14 did not indicate any changes in the *COL7A1* transcript, suggesting the cross-annealing was not significant.

Exon 73 is the largest exon within the *COL7A1* gene consisting of 201 nucleotides, and this presumably contributes to this exon having the highest number of recorded mutations in the DEB Register (accessed on 15 September 20) [25,26,27]. According to this database, 55 mutations are reported in exon 73, by far the greatest number compared to other exons, with 15 mutations reported in exon 74, the exon with the second highest number of mutations. However, the majority of mutations in exon 73 are missense mutations and correspond with the milder, dominant DEB. The spread of premature termination codons appears to be more even among various exons. The first clinical trial of an AO designed to skip *COL7A1* exon 73 was in Phase 1/Phase 2 (NCT03605069) [28]. In that study, a fully phosphorothioated 2′-OMe 21mer AO was administered topically to patients. While introducing large amounts of C7 via gene or replacement therapy into RDEB patients lacking this protein may induce an immunogenic response, producing C7 more slowly by AO-mediated exon skipping to remove exons, harboring null mutations is more likely to evade the immune response. 

We targeted exon 73, along with the 117 base exon 10, outside those encoding the collagenous region, to explore splice-switching capabilities using antisense oligonucleotides. Exon 73 was efficiently removed using 2′-OMe AO and PMO. However, this was not the case for exon 10 removal. We observed modest splice-switching after transfection of cells with exon 10 2′-OMe AOs, yet up to 100% removal of exon 10 when the equivalent sequences were administered to cells as PMOs. Irrespective of the 2′-OMe AO concentration transfected, we could not elicit exon skipping high enough to eliminate the full-length transcript. 

We were unable to directly compare 2′-OMe AOs and PMOs as they require different methods of transfection into cells in vitro: Lipofectamine 3000 was used to administer 2′-OMe AOs and electroporation by nucleofection was used to administer the PMOs. We tested the 2′-OMe AOs in fibroblasts at various concentrations via nucleofection. However, the cells did not survive. Possibly due to their toxicity [29], 2′-OMe AOs targeting *COL7A1* exons 10 and 73 tested at concentrations higher than 50 nM (approximately one femtomoles per cell) using Lipofectamine 3000 showed reduced transfection efficiency and a notable increase in cell death. The 150 µM concentration of PMO used in this study was nucleofected into a larger number of cells and hence the calculated number of molecules per cell (approximately 0.1 femtomoles) was roughly tenfold less compared to the highest 2′-OMe AO treatments. It is well recognized that PMOs have poor cellular/nuclear uptake due to their neutral backbone, and a much higher concentration is required to enter the cell. However, once inside, their neutral backbone elicits a higher binding affinity to RNA molecules when compared to 2′-OMe AOs [30,31,32]. 

Some 2′-OMe AOs targeting exon 73 showed consistent nonspecific exon skipping and inclusion of exons and introns, other than exon 73, and this effect was most pronounced when the AO impinged on an SRSF5 (SRp40) site. Of the seven bases corresponding to the predicted SRSF5 site (TTCCTGG), masking only two bases resulted in multiple splice isoforms, whereas shifting oligomer annealing one nucleotide upstream/downstream of the SRSF5 site eliminated these off-target effects and induced, specifically, transcripts missing exon 73. While these experiments were conducted in human fibroblasts, we do not expect the splicing pattern to differ much between fibroblasts and keratinocytes. However, as the transfection efficiency of keratinocytes is reduced compared to fibroblasts, the ratio of the different transcripts may fluctuate [11]. It has yet to be determined which cell type, fibroblast or keratinocyte, is best to target with AO intervention to restore C7 expression as both cell types produce C7 and are part of the epidermal basement membrane layers [33].

To further investigate the mechanism behind the unanticipated inclusion of intron 76 and combined removal of exons 73 and 74, we employed the approach described by Kessler et al. [13] to determine the removal order of *COL7A1* introns 72 to 76 through the use of strategically placed primer sets to amplify the pre-mRNA molecule. By this method, it was determined introns 72 and 75 were removed before introns 73 and 74, which were, in turn, removed before intron 76. According to the supraspliceosome model, the supraspliceosome acts as a multiprocessor machine that can concurrently splice four introns, albeit not necessarily in sequential order [34]. It would appear that the AOs targeting exon 73 for removal influence the pre-mRNA splicing of intron 76, since intron 76 is removed serially after intron 73 and 74. At 75 bases in length, intron 76 could easily be recognized as a coding domain in the exon-dense and compact *COL7A1* pre-mRNA, and AO-induced removal of earlier exons could easily compromise the pre-mRNA processing in unexpected ways. Furthermore, it should be noted that using an RT-PCR assay across exons 71 and 75 to monitor exon 73 skipping, as reported by Turczynski et al. [11] and Bornert et al. [28], would fail to detect the inclusion of intron 76. However, if intron removal order is the only mechanism responsible for the retention of intron 76, then any AO targeting exon 73 or 74 for removal should result in intron 76 retention, which we did not observe. Other influences, such as pre-mRNA secondary structure, AO sequence and chemistry, exonic splicing enhancer and silencer profiles, and cell type utilized, to name a few, are likely to contribute to aberrant splicing patterns. Additional studies are required to determine if the order of intron removal is valuable to decipher the mechanisms behind multiexon skipping and/or intron retention in the processing of the *COL7A1* pre-mRNA.

Analyzing the functionality and trimerization of C7 isoforms, encoded by transcripts lacking single exons after AO treatment, is the next logical step to be considered. An expression construct system could render it possible to facilitate strategies to assess systematic removal of exons and evaluate functionality of the resultant proteins [5]. However, to discriminate between dispensable exons and those necessary for functional anchoring of fibrils, development of 107 constructs is needed to test all in-frame exons. Even more constructs would be necessary to evaluate the consequences of removing blocks of exons. While case reports suggest that the skipping of mutated exons in some patients results in less severe DEB [35,36,37,38,39], not every in-frame exon has been characterized in this manner. Hence, prioritization of amenable targets should be determined by those RDEB individuals with milder than anticipated phenotypes.

To conclude, AO-mediated exon skipping is a highly promising treatment for RDEB. However, this approach does come with many challenges, including designing and optimizing AO candidates specific to the target exon, without causing mis-splicing events in other regions of this highly repetitive coding sequence. The order in which introns are removed from the pre-mRNA may play an important role in AO-mediated multiexon skipping and intron retention events in *COL7A1* pre-mRNA processing. However, the order of intron splicing may not be the only mechanism behind these aberrant splicing events, and additional studies are required to ultimately bring safe and sustainable drugs into the clinic for what will often be individual cases. 

## 4. Materials and Methods 

### 4.1. AO Design and Synthesis

Antisense oligomers were designed to anneal to the natural *COL7A1* exon acceptor or donor splice sites, or intraexonic splice enhancer (ESE) motifs predicted by ESEfinder 3.0 [12]. 2′-OMe AOs were synthesized by TriLink BioTechnologies (San Diego, CA, USA) or synthesized in-house on an Expedite 8909 nucleic acid synthesizer (Applied Biosystems, Melbourne, Australia) using the 1 µmol thioate synthesis protocol, as described previously [40]. Following incubation in ammonium hydroxide for 16 h at room temperature, the oligonucleotides were cleaved from the support and desalted under sterile conditions on NAP-10 columns (GE Healthcare, Sydney, Australia) according to manufacturer’s instructions. Oligomer sequences identified as inducing the most efficient exon skipping were subsequently purchased as PMOs (Gene Tools LLC, Philomath, OR, USA). Oligomer nomenclature is based on that described by Aung-Htut et al. [41], indicating the intron:exon, exon or exon:intron annealing coordinates in the *COL7A1* pre-mRNA. The AO oligomers targeting exons 10 and 73 are shown in Table 2.

### 4.2. Cell Propagation and Transfection

Primary dermal fibroblasts derived from a healthy volunteer after informed consent (Murdoch University Human Research Ethics Committee approval 2013/156, 25 October 2013; Murdoch University Human Research Ethics Committee approval 2017/101, 12 May 2017) were propagated in Dulbecco’s Modified Eagle’s Medium (DMEM) (Gibco; Thermo Fisher Scientific, Scoresby, Australia) supplemented with 15% fetal calf serum (FCS) (Serana, Bunbury, Australia) and 1% GlutaMax™-I (Gibco) at 37 °C in 5% CO_2_ atmosphere. Cells were seeded in 24-well plates (1.5 × 10^4^ cells/well) in DMEM supplemented with 10% FCS and cultured for 24 h prior to transfection. 

Fibroblasts were transfected with 2′-OMe AO-Lipofectamine 3000 (Thermo Fisher Scientific) lipoplexes in Opti-MEM (Gibco) according to the manufacturer’s instructions, at concentrations of 10, 25 and 50 nM in duplicate wells, and the cells were then incubated at 37 °C for 24 h before RNA extraction. Prior to nucleofection, PMO solutions were warmed for 5 min at 37 °C and nucleofection was performed using the P3 Primary Cell 4D-nucleofector X kit S (32 RCT) (Lonza, Mt Waverly, Australia) with CA 137 program setting. Approximately 3.5 × 10^5^ cells were used for nucleofection with 150 µM PMO in the 20 µl cuvette before plating into duplicate wells in a six-well plate. The cells were incubated for four days before harvesting. The standard control morpholino oligo (Gene Tools, LLC) that targets a human beta-globin intron mutation, was used as a negative control. All AO sequences are shown in Table 2. 

### 4.3. Molecular Analyses

After harvesting the cells, total RNA was extracted using a MagMax™ nucleic acid isolation kit (AM1830) (Thermo Fisher Scientific) according to manufacturer’s instructions and included the DNase treatment step. Molecular analyses were accomplished using two different systems optimized for different regions of *COL7A1* transcript. SuperScript™ III One-Step RT-PCR System with Platinum™ *Taq* DNA Polymerase (Thermo Fisher Scientific) was used to synthesize and amplify cDNA from 50 ng of total RNA in a single step. For regions with a high GC-content that are more difficult to amplify, SuperScript™ IV First-Strand Synthesis System and random hexamers (Thermo Fisher Scientific) were used to synthesize cDNA from harvested total RNA, and approximately 50 ng of cDNA was used as a template for PCR amplification using the TaKaRa LA Taq^®^ DNA Polymerase with GC Buffer II system (Takara Bio USA, Inc., Clayton, Australia). Genomic DNA from untreated healthy primary human fibroblasts was extracted using PureLink^®^ Genomic DNA mini kit (Thermo Fisher Scientific) according to the manufacturer’s instructions, and 15 ng was used as a template for PCR amplification. PCR systems, conditions and primers used to assess exon skipping efficiency across *COL7A1* are summarized in Table 3. Pairwise comparison of intron removal order, with primer combinations, is summarized in Table 4 and Figure 3. 

Amplified RT-PCR products were resolved on 2% agarose gels by electrophoresis in Tris-acetate ethylenediaminetetraacetic acid buffer and compared to a 100 bp DNA size standard (Geneworks, Adelaide, Australia). Relative exon skipping efficiency was estimated by densitometry on images captured by the Fusion FX system (Vilber Lourmat, Marne-la-Vallée, France) using Fusion-Capt software (version 17.03) and Bio-1D software (version 15.06a) for densitometry analysis. To identify RT-PCR products, the amplicons were first isolated by band stab [42] followed by template preparation using Diffinity RapidTip for PCR purification (Diffinity Genomics, Inc., West Henrietta, NY, USA) and DNA Sanger sequencing, performed by the Australian Genome Research Facility Ltd. (Nedlands, Australia).

### 4.4. In Silico Analyses

The Basic Local Alignment Search Tool (BLAST) [24] was used to compare amplicon sequences to the reference genomic and mRNA sequences (Accession numbers: NG_007065.1 and NM_000094.3, respectively). The GC-content and thermodynamic properties of the AOs were analyzed using Oligo Calc: Oligonucleotide Properties Calculator [43].

## Figures and Tables

**Figure 1 ijms-21-07705-f001:**
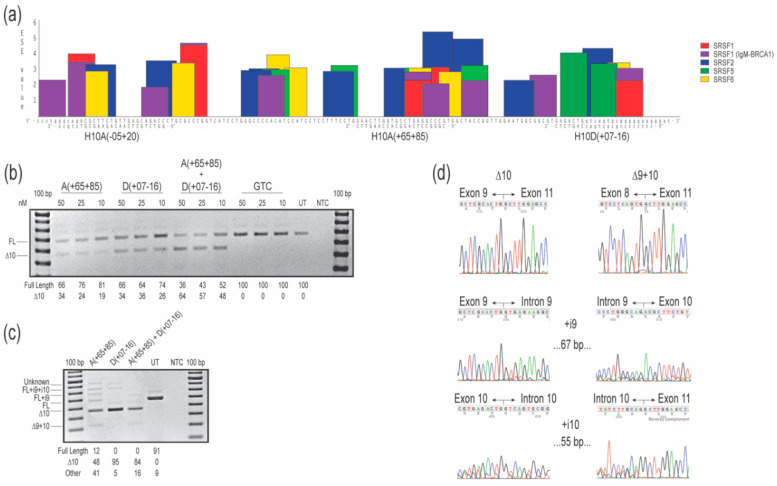
Evaluation of splice-switching antisense oligomers (AOs) targeting *COL7A1* exon 10 in healthy human fibroblasts. (**a**) Locations of exon splice enhancer (ESE) motifs predicted by ESEfinder 3.0 [12] and AO annealing sites targeted to remove exon 10. Relative predicted splice factor binding site scores are indicated on the *y*-axis. The color code indicates the predicted binding sites for the different serine and arginine rich splicing factors (SRSF). Exonic sequences are shown in upper case letters and intronic sequences are depicted in lower case. (**b**) Reverse transcription polymerase chain reaction (RT-PCR) analysis of *COL7A1* transcripts after transfection with 2′-*O*-methyl modified bases on a phosphorothioate backbone (2′-OMe) AOs. Transfection concentrations are indicated above the gel image. (**c**) Reverse transcription PCR analysis of *COL7A1* transcripts after phosphorodiamidate morpholino oligomers (PMOs) were nucleofected into the cells at a concentration of 150 µM in the cuvette. Relative abundance (%) of amplicons are shown under each gel image. Data are representative of two independent experiments. (**d**) Sanger sequencing data showing the different amplicons produced after treatment with AOs in (**c**). The gels were cropped for presentation. Full gel images are presented in Appendix A. GTC, Gene Tools control; NTC, no template control; UT, untreated; bp, base pairs; FL, full-length amplicon; nM, nanomolar; i, intron.

**Figure 2 ijms-21-07705-f002:**
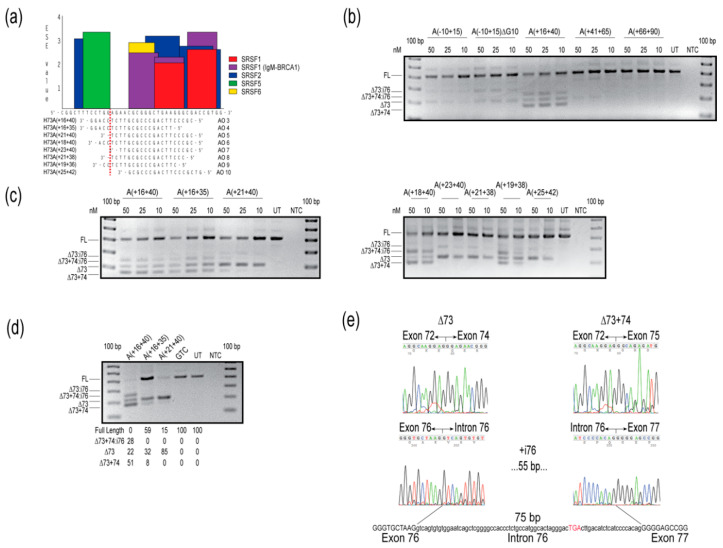
Evaluation of splice-switching antisense oligomers (AOs) targeting *COL7A1* exon 73 in healthy human fibroblasts. (**a**) Locations of AOs targeted to remove *COL7A1* exon 73, designed by micro-walking across the *COL7A1* H73(+16+40) annealing site, and the exon splice enhancer (ESE) motifs, predicted by ESEfinder 3.0 [12]. Relative splice factor binding scores, as predicted by ESEfinder 3.0 are indicated on the *y*-axis. The color code indicates the putative binding sites for serine and arginine rich splicing factors (SRSFs). (**b**) Reverse transcription polymerase chain reaction (RT-PCR) analysis of *COL7A1* transcripts after transfection with 2′-*O*-methyl modified bases on a phosphorothioate backbone (2′-OMe) AOs, at concentrations indicated above the gel image. (**c**) Reverse transcription PCR analysis of *COL7A1* transcripts after transfection with overlapping 2′-OMe AOs, at concentrations indicated above the gel image. (**d**) Reverse transcription PCR analysis of *COL7A1* transcripts after nucleofection with PMOs at a concentration of 150 µM in the cuvette, with relative abundance of amplicons (as percentage) shown below the gel image. Data are representative of at least two independent experiments. (**e**) Sanger sequencing data identifies the different amplicons resulting from AO treatment. Intronic sequence is represented in lower case, exonic sequence as upper case text. The premature termination codon in intron 76 is indicated by red upper case text. The gels were cropped for presentation. Full gel images are presented in Appendix A. GTC, Gene Tools control; NTC, no template control; UT, untreated; bp, base pairs; FL, full-length amplicon; nM, nanomolar; i, intron.

**Figure 3 ijms-21-07705-f003:**
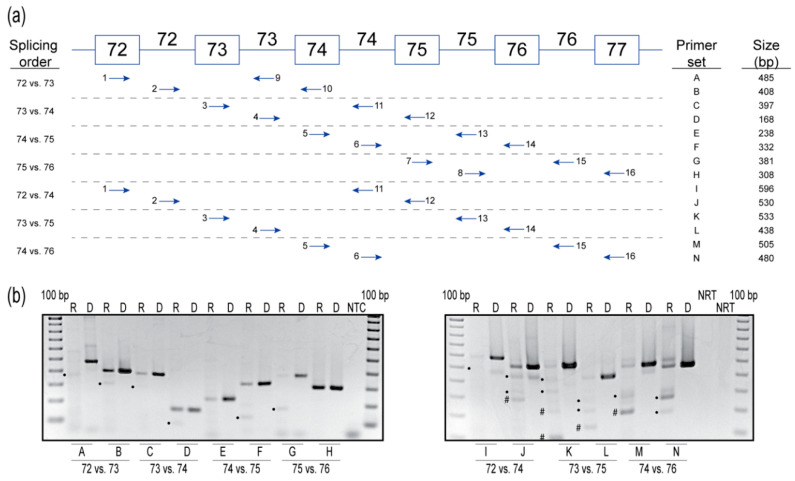
(**a**) Primer sets used to determine order of intron removal. Exons are indicated by boxes, and introns indicated by lines connecting the boxes. Arrows represent forward and reverse primers located either within an exon or intron. (**b**) Reverse transcription polymerase chain reaction analysis of the order of intron removal between exons 72 and 77. For each primer set, a sample of genomic DNA (D) was amplified to show the expected unspliced product size, along with an RNA (R) sample from healthy human untreated fibroblasts. Expected amplicons with introns removed are indicated by a dot (●). Unknown amplicons are indicated by a hash (#). Data are representative of two independent experiments. The gels were cropped for presentation. Full gel images are presented in Appendix A. NTC, no template control; NRT, no reverse transcriptase control; bp, base pairs; R, RNA; D, genomic DNA.

**Table 1 ijms-21-07705-t001:** Summary splicing analysis strategies and splicing order for paired introns in *COL7A1* pre-mRNA, from introns 72 to 76.

Intron Pair	Primer Set	Intron Spliced	Spliced Product Expected Sizes (bp)	Order of Splicing
72 vs. 73	A	72	399	72 generally removed before 73
	B	73 ^1^	312	
73 vs. 74	C	73 ^1^	301	No preference 73 = 74
	D	74 ^1^	88	
74 vs. 75	E	None	158	75 removed before 74
	F	75	134	
75 vs. 76	G	75	183	75 removed before 76
	H	None	233	
72 vs. 74	I	72 ^1^	510; 500; 414	Inconclusive
	J	?	450; 434; 354	
73 vs. 75	K	73 ^1^ or 74 ^1^, 73+74 ^1^	458; 442; 362	75 removed before 73
	L	74, 75, 74+75	413; 295; 215	
74 vs. 76	M	74, 75, 74+75	429; 311; 231	74 and 75 removed before 76
	N	75, 75+76 ^1^	405; 282; 207	

^1^ Indicates a minor product.

**Table 2 ijms-21-07705-t002:** Splice-switching antisense oligomers (AOs) designed and evaluated to excise *COL7A1* exons 10 and 73. 2′-*O*-methyl modified bases on a phosphorothioate backbone (2′-OMe) AOs are synthesized with uracil, while the corresponding phosphorodiamidate morpholino oligomers (PMOs) are synthesized with thymidine, indicated with an asterisk (*). The standard control morpholino was purchased from Gene Tools LLC (Philomath, OR, USA).

AO Nomenclature and Annealing Coordinates	Sequence (5′–3′)
H10A(-05+20)	GGUCUGCUCAACAGAAGCGUCUGCC
H10A(+65+85) *	CGGGCCUCAGGCACCAAGUUC
H10D(+07-16) *	CUUCCCCCGCACUGACCAGUCUC
H73A(-10+15)	AAAGCCGAUGGGGCCCUGCAGGAGU
H73A(-03+17)	GGAAAGCCGAUGGGGCCCUG
H73A(+01+20)	CCAGGAAAGCCGAUGGGGCC
H73A(+04+23)	UCUCCAGGAAAGCCGAUGGG
H73A(+07+26)	CGUUCUCCAGGAAAGCCGAU
H73A(+10+29)	CCGCGUUCUCCAGGAAAGCC
H73A(+13+32)	AGCCCGCGUUCUCCAGGAAA
H73A(+16+35) *	UUCAGCCCGCGUUCUCCAGG
H73A(+16+40) *	CGCCCUUCAGCCCGCGUUCUCCAGG
H73A(+18+40)	CGCCCUUCAGCCCGCGUUCUCCA
H73A(+19+36)	CUUCAGCCCGCGUUCUCC
H73A(+21+38)	CCCUUCAGCCCGCGUUCU
H73A(+21+40) *	CGCCCUUCAGCCCGCGUUCU
H73A(+23+40)	CGCCCUUCAGCCCGCGUU
H73A(+25+40)	CGCCCUUCAGCCCGCG
H73A(+25+42)	GUCGCCCUUCAGCCCGCG
H73A(+27+40)	CGCCCUUCAGCCCG
H73A(+41+65)	CCCUGAGGGCCAGGGUCUCCACGGU
H73A(+66+90)	CUCCCCAAGGGCCAGACCAGGUGGC
H73A(+91+115)	GGCCGGAAGGCCCGGGGGGGCCCCU
H73A(+116+140)	CCAGGCUUUCCAGGCUCCCCGGCAA
H73A(+141+165)	AGCCCUGCCUGGGAGCCCGGGAAUA
H73A(+166+190)	GCCUUCCUGCCUCUCCCACACCCCC
H73D(+11-14)	GCCCCCAGCCUCACCCUCUCUCCUG
Standard control morpholino *	CCUCUUACCUCAGUUACAAUUUAUA

**Table 3 ijms-21-07705-t003:** Primer pairs and amplification conditions used to assess the efficiency of *COL7A1* exon 10 and 73 skipping.

Primer Orientation	Sequence (5′–3′)	Length (bp)	PCR System	Cycling Conditions
Exon 8F	AACTGACCATCCAGAATACC	698	SSIII One-Step	55 °C (30 min) and 94 °C (2 min); 30 cycles of 94 °C (30 s), 60 °C (1 min) and 68 °C (2 min)
Exon 13R	GTCATCCAAGTCGAATGCT			
Exon 72F	AGATCGTGGAGACCTGGGATG	521	SSIV TaKaRa GC II	94 °C (1 min); 35 cycles of 94 °C (30 s), 60 °C (1 min) and 72 °C (2 min)
Exon 77R	CCTGTCTCCTTTGGGACCTTG			

**Table 4 ijms-21-07705-t004:** Primers used to study the removal order of *COL7A1* introns 72 to 76.

Primer Number	Orientation	Region	Sequence (5′–3′)
1	Forward	Exon 72	AGATCGTGGAGACCTGGGATG
2	Forward	Intron 72	TGAGCAGAAGTGGCTCAGTG
3	Forward	Exon 73	ATCGGCTTTCCTGGAGAACG
4	Forward	Intron 73	ACCCGCTATTTGCATTTCAG
5	Forward	Exon 74	AACGGGGAGAGAAAGGAGAA
6	Forward	Intron 74	GCTGCCACCCCATTTTCTTG
7	Forward	Exon 75	CCTCCTGGACTCCCTGGAAC
8	Forward	Intron 75	GTGTGTGCCATAACCCTGGA
9	Reverse	Intron 73	ATGCAAATAGCGGGTGAGGG
10	Reverse	Exon 74	CGTTCTCCTTTCTCTCCCCG
11	Reverse	Intron 74	CAAGAAAATGGGGTGGCAGC
12	Reverse	Exon 75	GTTCCAGGGAGTCCAGGAG
13	Reverse	Intron 75	ACCAAGCTAAGGGTGGCTTC
14	Reverse	Exon 76	TGTTCTCCAGAGAGTCCAGG
15	Reverse	Intron 76	GATGTCAAGTCAGTCCCTAGTGC
16	Reverse	Exon 77	CCTGTCTCCTTTGGGACCTTG

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
