# Peer review of "Nonsequential Splicing Events Alter Antisense-Mediated Exon Skipping Outcome in COL7A1"

_ijms, 2020, doi:10.3390/ijms21207705_

Round 1

Reviewer 1 Report

This is weel-written and well-designed study addressing exon skipping for COL7A1, the gene which is mutated in dystrophic epidermolysis bullosa.

The authors chose to target exons 10 and 73 with two chemistries, 2-ortho methy and phosphorodiamidate. They find that antisense oligomers induce different splicing patterns, including off target effects.

Minor comments:

  1. The references for the common occurence of mutations in exon 73 are very old. Since then hundreds of patients have been tested. In fact, exon 73 harbours most dominant mutations that usually yield mild phenotypes, but not most PTC mutations. This fact does not change the value of the manuscript, but such old literature has unfortunately induce confusion in the field.
  2. In this study, fibroblasts have been used. Is the splicing pattern expected to be the same in keratinocytes?
  3. The authors state that patients with two PTC mutations should be treated first; is there any concern about immunogenicity of the newly formed collagen VII?
  4. Can the authors comment on the clinical trial QR-313 employing AONs for exon 73 skipping?

Author Response

Reviewer 1:

Comments and suggestions for the authors: This is well-written and well-designed study addressing exon skipping for COL7A1, the gene which is mutated in dystrophic epidermolysis bullosa.
The authors chose to target exons 10 and 73 with two chemistries, 2-ortho methyl and phosphorodiamidate. They find that antisense oligomers induce different splicing patterns, including off target effects.
Response: Thank you for your time reviewing our manuscript. We value your comments and suggestions and have sought to respond to each one as requested.

Comment 1: The references for the common occurrence of mutations in exon 73 are very old. Since then hundreds of patients have been tested. In fact, exon 73 harbours most dominant mutations that usually yield mild phenotypes, but not most PTC mutations. This fact does not change the value of the manuscript, but such old literature has unfortunately induce confusion in the field.
Response 1: Thank you for your comments. We have now included an evaluation of the recent DEB register (The international database of dystrophic epidermolysis bullosa patients and COL7A1 mutations) in relation to overall and PTC mutations. (page 9 second paragraph).

Comment 2: In this study, fibroblasts have been used. Is the splicing pattern expected to be the same in keratinocytes?
Response 2: We intend to perform similar studies in keratinocytes, but due to current expected delivery times, the newly purchased cells have not arrived. Therefore, our study has only been conducted in normal human fibroblasts. We have now included a discussion on the possibility of different splicing isoforms and the proportion of the isoforms in keratinocytes compared to fibroblasts starting on page 9 paragraph 5. “While these experiments
were conducted in human fibroblasts, we do not expect the splicing pattern to differ much between fibroblasts and keratinocytes. However, as the transfection efficiency of keratinocytes is reduced compared to fibroblasts, the ratio of the different transcripts may fluctuate. It has yet to be determined which cell type, fibroblast or keratinocyte, is best to target with AO intervention to restore C7 expression as both cell types produce C7 and are part of the epidermal basement membrane layers.”

Comment 3: The authors state that patients with two PTC mutations should be treated first; is there any concern about immunogenicity of the newly formed collagen VII?
Response 3: Immunogenicity may be a concern if we administer/apply C7 protein to a patient with no previous C7 protein production. However, since we are stimulating the host’s cells to produce a form of the C7 protein, it is less likely to be immunogenic. Additionally, we have included the statement to page 9 paragraph 2) “While introducing large amounts of C7 via gene or replacement therapy into RDEB patients lacking this protein may induce an immunogenic response, producing C7 more slowly by AO-mediated exon skipping to remove exons harboring null mutations is more likely to evade the immune response.”

Comment 4: Can the authors comment on the clinical trial QR-313 employing AONs for exon 73 skipping?
Response 4: We have referred to the QR-313 clinical trial (page 9 paragraph 2), however, our knowledge was limited due to the lack of peer-reviewed publications available at the time of submission. A pre-print has just become available (Bornert et al. 2020), which presents the preclinical data. We have now included this reference into the manuscript. 

Reviewer 2 Report

General comments:
The presented study investigated exon skipping therapy for the devastating skin blistering disease dystrophic epidermolysis bullosa. The authors investigated exon skipping as potential therapeutic approach for mutations located in exon 73 and exon 10 of the COL7A1 gene. During these experiments, the authors stumbled upon the interesting phenomenon of non-sequentially spliced introns. Which is crucial information for exon skipping therapy design.

These findings are interesting, however, the novelty of these findings could be questioned. Especially, due to the main message of the manuscript. The authors suggest to design analysis tools spanning multiple exons surrounding the targeted exon. This is common practice.

However, the data shown of non-sequential and alternative splicing of the pre-mRNA are very interesting findings. I would strongly recommend to re-aim the manuscript towards these findings. Conclude that the non-sequential splicing nature of certain introns exclude corresponding exons from being targeted by antisense oligonucleotide-mediated exon skipping. Or expand the knowledge of splicing order of all (relevant to exon skipping) COL7A1 pre-mRNA introns.

In conclusion, the findings are valuable. In my opinion, the value of the data would be highly elevated if presented with a different message.

Specific comments:

  1. The comparison between the two chemistries made in the introduction of the manuscript (page 2 line 76-81) is speculative. The comparison is made with extreme variables in an in vitro setting. Different concentrations, and more importantly different transfection methods are compared. It is well known that the bioavailability and efficiency between the PMO and 2'-OMePS differ, and that the in vitro results cannot be extrapolated to the clinic. This raises the concern of potential conflict of interest as the PMO chemistry is a platform of Sarepta Therapeutics.
  2. In the introduction both dystrophic epidermolysis bulls and epidermolysis bulls dystrophica are used. This should be consistent.
  3. COL7A1 and COL7A1 are used as name for gene and protein, respectively. The protein name, type VII collagen, or other abbreviation (e.g. C7 or col7) should be used, as only non-italic for protein is confusing.
  4. Page 2 line 69: abbreviation AO should be introduced a priori
  5. It is unclear why two different transfection methods and extreme difference in concentration are used. This should be further explained.
  6. Although not essential for this study in particular. The quantification of exon skipping should be performed real-time using qPCR.

Author Response

Reviewer 2:

Comments and suggestions for the authors: The presented study investigated exon skipping therapy for the devastating skin blistering disease dystrophic epidermolysis bullosa. The authors investigated exon skipping as potential therapeutic approach for mutations located in exon 73 and exon 10 of the COL7A1 gene. During these experiments, the authors stumbled upon the interesting phenomenon of non-sequentially spliced introns. Which is crucial information for exon skipping therapy design.

These findings are interesting, however, the novelty of these findings could be questioned. Especially, due to the main message of the manuscript. The authors suggest to design analysis tools spanning multiple exons surrounding the targeted exon. This is common practice.

However, the data shown of non-sequential and alternative splicing of the pre-mRNA are very interesting findings. I would strongly recommend to re-aim the manuscript towards these findings. Conclude that the non-sequential splicing nature of certain introns exclude corresponding exons from being targeted by antisense oligonucleotide- mediated exon skipping. Or expand the knowledge of splicing order of all (relevant to exon skipping) COL7A1 pre- mRNA introns.

In conclusion, the findings are valuable. In my opinion, the value of the data would be highly elevated if presented with a different message.

Response: Thank you for your time reviewing our manuscript. Although it seems intuitional to design the analysis tools spanning multiple exons and this practise is regarded as a “common practice”, we have referenced two articles (Turczynski et al. 2016; Bornert et al. 2020) in particular where the authors, using their described methods for analysis, would have overlooked the mis-splicing events we observed. While scientists with extensive experience in developing splice-switching therapeutics should understand the value of an assay that examines multiple exons either side of the targeted exon, the use of antisense oligonucleotides is becoming more widespread and this knowledge or experience may be lacking. Honesty, this is our first observation of a mis-splicing event this far away from a targeted exon, and we were fortunate to discover it. However, we do agree that appropriate assay design should not be the key message of the manuscript and have adjusted the focus slightly. We value your comments and suggestions and have sought to respond to each one below as requested.

Comment 1: The comparison between the two chemistries made in the introduction of the manuscript (page 2 line 76-81)  is  speculative.  The  comparison  is  made  with  extreme  variables  in  an in  vitro setting.  Different concentrations, and more importantly different transfection methods are compared. It is well known that the bioavailability and efficiency between the PMO and 2'-OMePS differ, and that the in vitro results cannot be extrapolated to the clinic. This raises the concern of potential conflict of interest as the PMO chemistry is a platform of Sarepta Therapeutics.

Response 1: As we have now changed the message of the manuscript at the suggestion of the reviewer to focus more on the non-sequential intron splicing order observed in COL7A1, we have removed the comment of clinical relevance of the two chemistries from page 2 paragraph 5.

Comment 2: In the introduction both dystrophic epidermolysis bulls and epidermolysis bulls dystrophica are used. This should be consistent.

Response 2: The terminology consistency has been rectified and we have changed “epidermolysis bullosa dystrophica” (page 2 paragraph 1) to “dystrophic epidermolysis bullosa” as requested.

Comment 3: COL7A1 and COL7A1 are used as name for gene and protein, respectively. The protein name, type VII collagen, or other abbreviation (e.g. C7 or col7) should be used, as only non-italic for protein is confusing.

Response 3: We have now changed all COL7A1 referring to the protein to C7, as requested.

Comment 4: Page 2 line 69: abbreviation AO should be introduced a priori

Response 4: We have introduced the abbreviation “AO” when it was first mentioned, page 2 paragraph 2.

Comment 5: It is unclear why two different transfection methods and extreme difference in concentration are used. This should be further explained.

Response 5: We have now included some remarks on the different transfection methods on page 9 paragraph 4. In particular, we have included “We were unable to directly compare 2′-OMe AOs and PMOs as they require different methods of transfection into cells in vitro: lipid nanoparticles were used to administer 2′-OMe AOs and electroporation by nucleofection was used to administer the PMOs” and “We tested the 2′-OMe AOs at various concentrations via nucleofection, however, the cells did not survive.” We have also added the number of moles/cell to more accurately represent the comparison between PMO and 2’-OMe AO transfection and have added the sentence “The 150 µM concentration of PMO used in this study was nucleofected into a much larger number of cells and hence calculated to be around 10 times less molecules per cell compared to the highest 2′-OMe AO treatments.”

Comment 6: Although not essential for this study in particular. The quantification of exon skipping should be performed real-time using qPCR.

Response 6: We agree that the use of qPCR is valuable to determine the efficiency of exon skipping, however we decided not to perform qPCR as RT-PCR analysis in our opinion illustrated clearly the differences in the splice- switching patterns observed.

Round 2

Reviewer 2 Report

Start quote"

These findings are interesting, however, the novelty of these findings could be questioned. Especially, due to the main message of the manuscript. The authors suggest to design analysis tools spanning multiple exons surrounding the targeted exon. This is common practice.

However, the data shown of non-sequential and alternative splicing of the pre-mRNA are very interesting findings. I would strongly recommend to re-aim the manuscript towards these findings. Conclude that the non-sequential splicing nature of certain introns exclude corresponding exons from being targeted by antisense oligonucleotide- mediated exon skipping. Or expand the knowledge of splicing order of all (relevant to exon skipping) COL7A1 pre- mRNA introns.

In conclusion, the findings are valuable. In my opinion, the value of the data would be highly elevated if presented with a different message.

Response: Thank you for your time reviewing our manuscript. Although it seems intuitional to design the analysis tools spanning multiple exons and this practise is regarded as a “common practice”, we have referenced two articles (Turczynski et al. 2016; Bornert et al. 2020) in particular where the authors, using their described methods for analysis, would have overlooked the mis-splicing events we observed. While scientists with extensive experience in developing splice-switching therapeutics should understand the value of an assay that examines multiple exons either side of the targeted exon, the use of antisense oligonucleotides is becoming more widespread and this knowledge or experience may be lacking. Honesty, this is our first observation of a mis-splicing event this far away from a targeted exon, and we were fortunate to discover it. However, we do agree that appropriate assay design should not be the key message of the manuscript and have adjusted the focus slightly.

" End quote

I appreciate that the authors agree that the key message of the manuscript should not be the appropriate assay design. However, this message of the manuscript has not been changed. For example, the concluding sentence of the abstract, "We found that the non-sequential splicing of certain introns may alter pre-mRNA processing during antisense oligomer-mediated exon skipping, and therefore recommend that any assay designed to detect antisense oligomer-induced skipping of collagen gene exons should encompass several exons flanking the target.",  which literally states assay design.

Additionally, the concluding paragraph: "To conclude, exon skipping-mediated therapy is a highly promising treatment for RDEB. However, this approach does come with many challenges, including designing and optimizing AO candidates specific to the target, without causing mis-splicing events in other regions of this highly repetitive coding sequence, utilizing experimental assays able to detect possible off-target effects while monitoring therapeutic benefits, and ultimately bringing safe and sustainable drugs into the clinic for what will often be individual cases."

I strongly recommend to revisit the aim and conclusions of the study. 

Additionally, the authors state in the discussion "However, to discriminate between dispensable exons and those necessary for functional anchoring of fibrils, development of 107 constructs is needed to test all in-frame exons. Even more constructs would be necessary to evaluate the consequences of removing blocks of exons. While case reports suggest that the skipping of mutated exons in some patients results in less severe DEB [38-42], not every in-frame exon has been characterized in this manner. Hence, prioritization of amenable targets should be determined by those RDEB individuals with milder than anticipated phenotypes.".
I agree with the authors that expansion of the data on splicing order and naturally occurring alternative splicing variants would be of great interest. Therefore, I encourage the authors to perform additional experiments to investigate the splicing of all therapeutically relevant exons of COL7A1.

Author Response

Second response to reviewer 2

I appreciate that the authors agree that the key message of the manuscript should not be the appropriate assay design. However, this message of the manuscript has not been changed. For example, the concluding sentence of the abstract, "We found that the non-sequential splicing of certain introns may alter pre-mRNA processing during antisense oligomer-mediated exon skipping, and therefore recommend that any assay designed to detect antisense oligomer-induced skipping of collagen gene exons should encompass several exons flanking the target.",  which literally states assay design (1).

(1).  Response: This concluding sentence has been altered to include (page 1; abstract): “… additional studies are required to determine if the order of intron removal influences multi-exon skipping and/or intron retention in the processing of the COL7A1 pre-mRNA.”

Additionally, the concluding paragraph: "To conclude, exon skipping-mediated therapy is a highly promising treatment for RDEB. However, this approach does come with many challenges, including designing and optimizing AO candidates specific to the target, without causing mis-splicing events in other regions of this highly repetitive coding sequence, utilizing experimental assays able to detect possible off-target effects while monitoring therapeutic benefits, and ultimately bringing safe and sustainable drugs into the clinic for what will often be individual cases."

I strongly recommend to revisit the aim and conclusions of the study (2).

(2).  Response: We have removed the comment referring to experimental assays and have added the following sentence (page 10; paragraph 3): “The order in which introns are removed from the pre-mRNA may play an important role in AO-mediated multi-exon skipping and intron retention events in COL7A1 pre-mRNA processing. However, the order of intron splicing may not be the only mechanism behind these aberrant splicing events and additional studies are required to ultimately bring safe and sustainable drugs into the clinic for what will often be individual cases.”

Additionally, the authors state in the discussion "However, to discriminate between dispensable exons and those necessary for functional anchoring of fibrils, development of 107 constructs is needed to test all in-frame exons. Even more constructs would be necessary to evaluate the consequences of removing blocks of exons. While case reports suggest that the skipping of mutated exons in some patients results in less severe DEB [38-42], not every in-frame exon has been characterized in this manner. Hence, prioritization of amenable targets should be determined by those RDEB individuals with milder than anticipated phenotypes.".
I agree with the authors that expansion of the data on splicing order and naturally occurring alternative splicing variants would be of great interest. Therefore, I encourage the authors to perform additional experiments to investigate the splicing of all therapeutically relevant exons of COL7A1 (3).

(3).  Response: It is our plan to identify exons that may be removed from the COL7A1 mature mRNA without seriously compromising function. The order of intron splicing will be considered relevant when “unexpected” events are encountered during the AO intervention. Unexpected events would include multiple exon skipping, intron retention and also exons that are very efficiently skipped or very difficult to dislodge. However, it would take months to perform the intron removal order studies for every intron in COL7A1. Since there are no obvious mutation hotspots in COL7A1 for RDEB, most of the in-frame exons encoding the collagenous domain could be therapeutically relevant targets. Although intron removal order may influence multi-exon skipping and intron retention, we are confident that it is not the sole mechanism and further studies are required. The first author, Mrs Kristin Ham has just begun her PhD, and these future studies will form a significant portion of her project.

We have included the statement (page 10 paragraph 1): “However, if intron removal order is the only mechanism responsible for the retention of intron 76, then any AO targeting exon 73 or 74 for removal should result in intron 76 retention, which we did not observe. Other influences, such as pre-mRNA secondary structure, AO sequence and chemistry, exonic splicing enhancer and silencer profiles, and cell type utilized, to name a few, are likely to contribute to aberrant splicing patterns. Additional studies are required to determine if the order of intron removal is valuable to decipher the mechanisms behind multi-exon skipping and/or intron retention in the processing of the COL7A1 pre-mRNA.”

For verification, the changes mentioned have been highlighted in yellow in the attached document

Round 3

Reviewer 2 Report

To the authors,

The issue regarding novelty remains. However, as I stated previously, the quality of the research is sound, and the results are presented clearly.

Minor suggestion regarding the title. To keep consistency with the text (see abstract):  "Non-sequential splicing events alter antisense-mediated exon skipping outcome in COL7A1"

Kind regards,